# Gene networks that compensate for crosstalk with crosstalk

Isaak E. Müller[1,2,8], Jacob R. Rubens[1,2,8], Tomi Jun [1,3,7], Daniel Graham [4,5], Ramnik Xavier [4,5,6] & Timothy K. Lu[1,2]

Crosstalk is a major challenge to engineering sophisticated synthetic gene networks. A common approach is to insulate signal-transduction pathways by minimizing molecular-level crosstalk between endogenous and synthetic genetic components, but this strategy can be difficult to apply in the context of complex, natural gene networks and unknown interactions. Here, we show that synthetic gene networks can be engineered to compensate for crosstalk by integrating pathway signals, rather than by pathway insulation. We demonstrate this principle using reactive oxygen species (ROS)-responsive gene circuits in *Escherichia coli* that exhibit concentration-dependent crosstalk with non-cognate ROS. We quantitatively map the degree of crosstalk and design gene circuits that introduce compensatory crosstalk at the gene network level. The resulting gene network exhibits reduced crosstalk in the sensing of the two different ROS. Our results suggest that simple network motifs that compensate for pathway crosstalk can be used by biological networks to accurately interpret environmental signals.

---

[1] Synthetic Biology Group, MIT Synthetic Biology Center, Research Laboratory of Electronics, Department of Electrical Engineering & Computer Science, Massachusetts Institute of Technology, Cambridge, MA 02139, USA. [2] Microbiology Program, Massachusetts Institute of Technology, Cambridge, MA 02139, USA. [3] Harvard-MIT Health Sciences and Technology Program, Cambridge, MA 02139, USA. [4] Program in Medical and Population Genetics, Broad Institute, Cambridge, MA 02142, USA. [5] Gastrointestinal Unit and Center for the Study of Inflammatory Bowel Disease, Massachusetts General Hospital, Boston, MA 02114, USA. [6] Center for Computational and Integrative Biology, Massachusetts General Hospital, Boston, MA 02114, USA. [7] Present address: Division of Hematology and Medical Oncology, Mount Sinai Hospital, New York, NY 10029, USA. [8] These authors contributed equally: Isaak E. Müller, Jacob R. Rubens. Correspondence and requests for materials should be addressed to T.K.L. (email: timlu@mit.edu)

Living cells use sophisticated signaling networks to assess the variable and complex environments they encounter. These networks have evolved insulated molecular-level signal-transduction[1,2] and network-level signal integration[3–6] mechanisms that enable them to implement robust and specific behavior with minimal unintended pathway crosstalk. For example, two-component systems, typically comprised of a sensor histidine kinase and a cognate response regulator, are the primary mode for bacteria to sense and respond to changes in the environment[7]. Despite high abundance of two-component systems in bacterial genomes and high sequence and structural similarity, signaling pathways are remarkably insulated. This specificity stems partially from a small number of amino acid residues at the site of histidine kinase and response regulator interaction[1], but cells have developed additional mechanisms that reduce crosstalk from a histidine kinase to a non-cognate response regulator. Many histidine kinases act both as kinase and phosphatase, thereby reversing phosphorylation of the response regulator by a non-cognate histidine kinase in the absence of input signal[8]. Similarly, response regulator competition can prevent crosstalk, as a histidine kinase exhibits kinetic preference to its cognate response regulator[9] and protein levels of response regulators are generally higher than histidine kinase levels[10]. While crosstalk in two-component system signaling pathways often has a negative impact on host fitness, bacteria also contain complex regulatory systems that integrate an array of inputs into one output, for example in the sporulation process of Bacillus subtilis[11]. In these natural systems, crosstalk is thus mitigated at the molecular-level through protein-protein interaction and reversible enzymatic reactions. Yet, crosstalk is a key challenge in the design of synthetic signaling and gene networks[12,13], as even the most commonly used and well-understood components exhibit undesirable interference[14,15], including when ported across kingdoms[16]. In attempts to minimize crosstalk, synthetic biologists have implemented molecular-level insulation by using inputs that are not naturally sensed by host cells[17], by knocking out endogenous genes[18,19], and by mutating and screening for orthogonal genetic components[20–23]. While these strategies have enabled the construction of moderately sized gene networks[24,25], it is an open question as to whether the insulation approach alone will scale to significantly more complex gene networks or will be adaptable to inputs that are sensed by or interface with host cell signaling networks. This challenge may be especially evident in contexts where it is undesirable to modify endogenous DNA, when the endogenous sensing pathway is too complex or integral to the host cell to modulate, or when the source of crosstalk is unknown. Here, we describe a complementary strategy to address gene circuit crosstalk by leveraging network-level signal integration and introducing crosstalk-compensating gene circuits. The crosstalk-compensation circuits are designed without detailed knowledge regarding the underlying source of the crosstalk and do not require any manipulation of endogenous genes. The circuits are similar to interference-cancellation circuits found in electrical engineering, where the output from a crosstalk-sensitive sensor is adjusted with the output from a sensor that specifically senses the interfering input.

Our sensor circuits are designed to sense the concentration of reactive oxygen species (ROS). We find that when exposed to two ROS simultaneously, one of our initial ROS-sensors exhibits significant crosstalk and one does not. We quantify the amount of crosstalk and use this information to design a circuit that compensates for the crosstalk by using the signal from the sensor that specifically detects the interfering ROS. This results in circuits that exhibit reduced crosstalk in the sensing of the two different ROS.

## Results

**Analog gene circuits evaluated with the utility metric.** We first created gene circuits that measure the concentration of $H_2O_2$ based on the OxyR transcriptional activator[26] in E. coli BW25113[27]. A central goal of this study was to construct circuits in the context of natural regulatory networks and therefore we did not knockout endogenous genes or otherwise manipulate chromosomal DNA. To characterize $H_2O_2$–OxyR responsive promoters, we built open-loop (OL) gene circuits with OxyR constitutively expressed from a medium-copy plasmid (MCP) and *mCherry* expression controlled by various OxyR-activated promoters on a high-copy plasmid (HCP) (Fig. 1a). The purpose of constitutively expressed OxyR was to establish a minimal level of OxyR in the cell independent of chromosomally expressed OxyR, which autoregulates its own expression naturally[28]. The measured input-output transfer curves were fit to Hill functions (Fig. 1b), which were used to calculate the sensitivity[29] of each circuit (Supplementary Fig. 2). Sensing analog information is important when the sum or ratio of signals is used for decision making[30–32] or when graded responses to a signal are necessary[33–37]. An ideal analog sensor circuit should have a high output fold-induction across a wide input dynamic range in order to transmit information over a broad concentration of input levels to downstream gene circuits or exogenous detectors. Here, we quantify this behavior with a metric called "utility". We calculated the utility for each circuit by determining the relative input range and multiplying by the output fold-induction (Supplementary Note 1), which are both parameters previously described[38,39]. Utility equally scores circuits with the same relative input range and output fold-induction independent of absolute input concentrations or output gene expression levels (Supplementary Fig. 1).

For our experiments, we used a maximum extracellular $H_2O_2$ concentration of 1.20 mM since concentrations above this inhibited cell growth. The promoter from the small RNA oxyS (oxySp)[40] outperformed the other OxyR-regulated promoters that have been previously utilized as sensors[41,42]. OxySp had both the second highest output fold-induction (15.0×) and the highest relative input range (58.4×). The utility was calculated to be 876.0, which was higher than the utility of the katGp circuit (324.2) and ahpCp circuit (214.9) (Fig. 1c).

Tuning OxyR production enhanced the performance of the $H_2O_2$-OxyR-oxySp circuit. We increased *oxyR* expression in the OL circuit by constitutively producing it from a strong proD promoter[43] on a high-copy plasmid (HCP), which is in addition to the genomic *oxyR* (Fig. 1d). This increased the output fold-induction to 23.6× and the relative input range to 63.0× (Fig. 1f), resulting in a utility of 1486.8 (Fig. 1g). Inspired by previous work that found that positive feedback can expand the input dynamic range for activating transcription factors[30], we tested OxyR in a positive-feedback (PF) circuit by fusing mCherry to the carboxy terminus of OxyR and placing the composite gene under the control of oxySp (Fig. 1e). This circuit had a wider relative input range (72.5×) than the OL circuit but a lower output fold-induction (15.9×) (Fig. 1f). The utility for the PF circuit was 1152.8 (Fig. 1g).

In addition to $H_2O_2$ sensors, we created paraquat-sensing circuits based on the SoxR transcription factor, which is reported to respond to $O_2-$ and redox-cycling agents such as paraquat[26,44,45]. We built an OL circuit with SoxR constitutively expressed from a MCP and *mCherry* expression controlled by the pLsoxS promoter on a HCP (Fig. 2a). The pLsoxS promoter has a SoxR binding site from the *soxS* promoter fused to the lambda phage −35 and −10 promoter region[46]. We also built a PF circuit with a SoxR-mCherry fusion protein expressed by the pLsoxS promoter on a HCP (Fig. 2b). We found that the OL circuit had

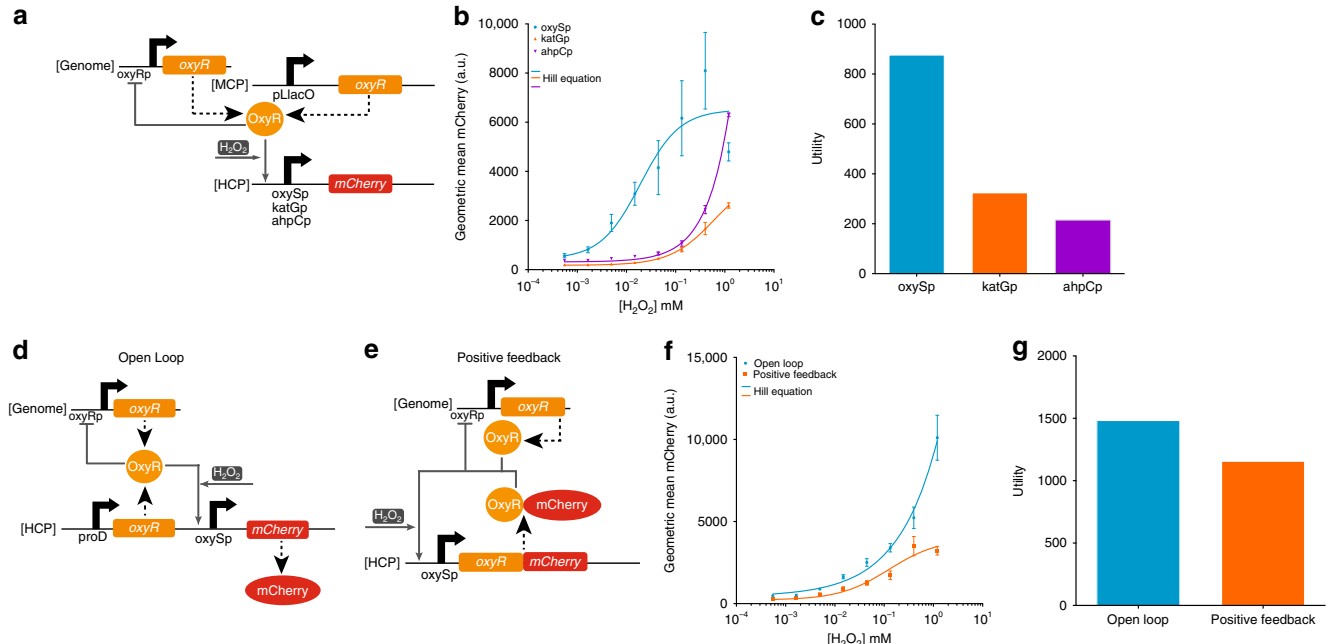

**Fig. 1** $H_2O_2$-sensing synthetic gene circuits. **a** The open-loop (OL) circuit used to test different $H_2O_2$-OxyR regulated promoters. OxyR is expressed from an unregulated constitutive pLlacO promoter on a medium-copy plasmid (MCP), and mCherry is expressed from different OxyR-activated promoters on a high-copy plasmid (HCP). OxyR activates *mCherry* expression in the presence of $H_2O_2$. OxyR is also expressed from the *E. coli* genome and negatively regulates its own production. Dashed large arrows represent protein production and solid small gray arrows represent transcriptional regulation. **b** Empirical $H_2O_2$-mCherry transfer functions for three different promoters (oxySp in blue, katGp in orange, ahpCp in purple). The lines are Hill equation fits to the raw data. Extracellular $H_2O_2$ concentrations above 1.20 mM were not tested due to toxicity. **c** The utility for the three different promoters calculated from the Hill functions in Fig. 1b. The circuit with the oxySp promoter has the highest utility. **d** The OL $H_2O_2$-OxyR-oxySp circuit. OxyR is expressed from the proD promoter and the oxySp promoter controls *mCherry* expression, both on a HCP. **e** The positive-feedback (PF) $H_2O_2$-OxyR-oxySp circuit. An OxyR-mCherry fusion protein positively regulates its own expression from the oxySp promoter on a HCP. In both the OL and PF circuits, OxyR is also expressed from the *E. coli* genome and negatively regulates its own expression. **f** The empirical $H_2O_2$-mCherry transfer function for the OL (blue) and PF (orange) $H_2O_2$-OxyR-oxySp circuits. The lines are Hill equation fits to the raw data. **g** The utility for the OL and PF $H_2O_2$-OxyR-oxySp circuits calculated using the Hill functions in Fig. 1f. The OL circuit has a higher utility. The errors (s.e.m.) are derived from three biological replicates and flow cytometry experiments, each involving 30,000 events. Source data are provided as a Source Data file

both a significantly larger output fold-induction (42.3× vs.10.2×) and relative input range (95.8× vs. 82.6×) than the PF circuit (Fig. 2c), resulting in a higher utility (4052.3 vs. 842.5) (Fig. 2d). However, a circuit with only the genomic copy of *soxR* had a higher utility than the OL circuit (4364.7) (Supplementary Fig. 3). Since SoxR binds to target promoters in the uninduced state and such transcription factors are often naturally found at low expression levels[47], we sought to determine whether a low constitutive level of *soxR* expression on top of the genomic *soxR* gene could yield better analog circuit performance. To do so, we transformed the OL circuit into an MG1655Pro *E. coli* strain that constitutively expresses the LacI repressor protein from the genome[48], allowing us to control *soxR* expression with the small molecule IPTG (Fig. 2e). Lower IPTG concentrations increased the output fold-induction, the relative input range, and the utility of the tested circuits (Fig. 2f, Supplementary Figs. 4 and 5). The lowest tested IPTG concentration maximized circuit utility to 11,620.0 (Fig. 2g).

**Crosstalk quantification in a dual-sensor strain.** We built a dual-sensor *E. coli* strain to explore how crosstalk in sensing paraquat and $H_2O_2$ impacts signal transduction in our sensor circuits. We were unable to assemble components of the optimized $H_2O_2$-sensing circuit (Fig. 1d) and paraquat-sensing circuit (Fig. 2e) together into an integrated DNA construct. Instead, the paraquat-sensing circuit was placed in an OL configuration with *soxR* constitutively expressed from a low copy plasmid (LCP) and

mCherry expression controlled by pLsoxS on a MCP (Fig. 3a). The $H_2O_2$-sensing circuit was put in an OL configuration and fully encoded on a HCP with sfGFP as an output. We found little crosstalk between paraquat and the $H_2O_2$-sensing circuit since *sfGFP* expression at any given $H_2O_2$ concentration was not considerably affected by paraquat (Fig. 3g). In contrast, the paraquat-sensing circuit was appreciably affected by $H_2O_2$ as *mCherry* expression at high paraquat concentrations was reduced by increasing concentrations of $H_2O_2$ (Fig. 3b).

To quantitate the amount of crosstalk in each sensing circuit, we first computed the raw error by subtracting the gene expression at every given paraquat and $H_2O_2$ concentration from the expected gene expression (i.e., the gene expression at the same concentration of only the cognate input, in the absence of the non-cognate input) (Supplementary Note 2). We then took the absolute value of the raw error to get the absolute error. We normalized the absolute error to expected gene expression to calculate the relative error, and summed the relative error over the entire range of input concentrations to get the total relative error. The total relative error was 23.5 for the paraquat-sensing circuit (Fig. 3h, Supplementary Fig. 7) and 12.9 for the $H_2O_2$-sensing circuit (Fig. 3i).

**Crosstalk compensation.** To address the crosstalk between $H_2O_2$ and the paraquat-sensing circuit, we designed synthetic circuits that introduce compensatory crosstalk. We used the raw error plot for the paraquat-sensing circuit as a blueprint for the ideal

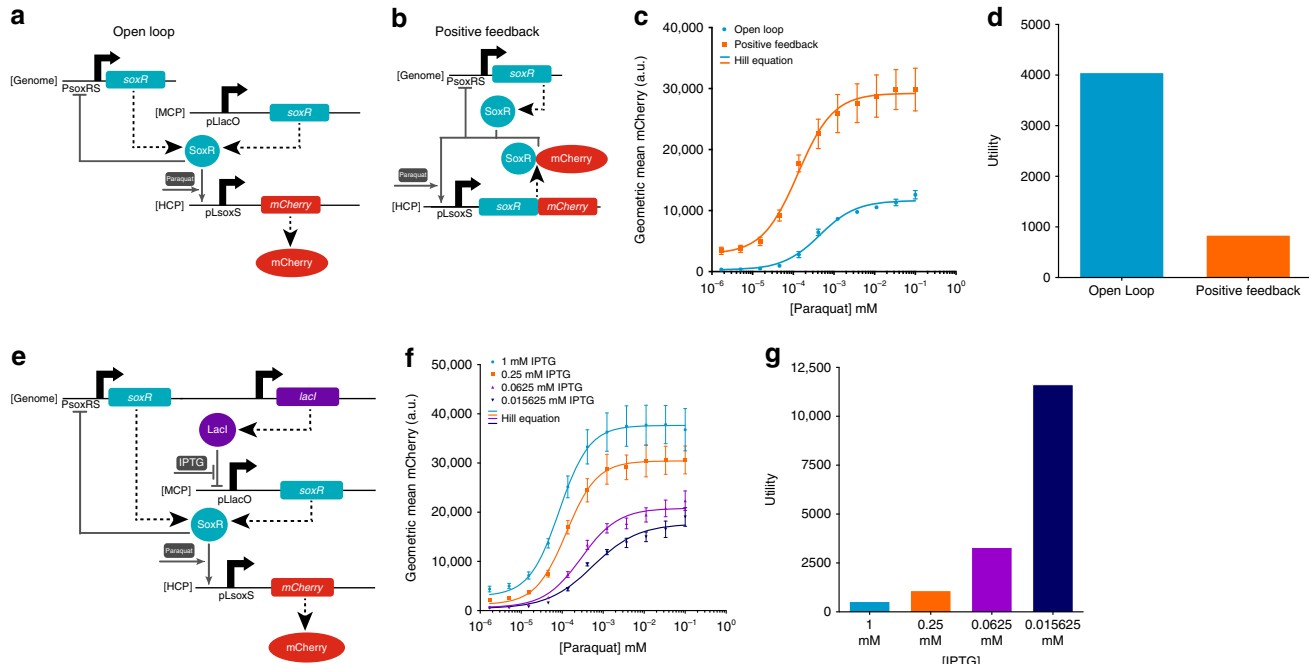

**Fig. 2** Paraquat-sensing synthetic gene circuits. **a** OL paraquat-SoxR-pLsoxS circuit. SoxR is expressed from an unregulated constitutive pLlacO promoter on a MCP, and mCherry is expressed from the pLsoxS promoter on a HCP. **b** The PF paraquat-SoxR-pLsoxS circuit. A SoxR-mCherry fusion protein positively regulates its own expression from the pLsoxS promoter on a HCP. In both the OL and PF circuits, SoxR is also expressed from the genome and negatively regulates its own expression[69]. Dashed large arrows represent protein production and solid small gray arrows represent transcriptional regulation. **c** The empirical paraquat-mCherry transfer function for the paraquat-SoxR-pLsoxS OL (blue) and PF (orange) circuits. The lines are Hill equation fits to the raw data. **d** The utility for the OL and PF circuits calculated from the Hill functions in Fig. 2c. The OL circuit has a higher utility. **e** The paraquat-SoxR-pLsoxS OL circuit in *E. coli* MG1655Pro cells. MG1655Pro constitutively expresses the *lacI* repressor, which represses the pLlacO promoter and thus *soxR* expression from the MCP. IPTG induces expression from the pLlacO promoter in this context. **f** The empirical paraquat-mCherry transfer functions for the paraquat-SoxR-pLsoxS OL circuits at different IPTG concentrations in MG1655Pro. **g** The utility calculated from the Hill functions in Fig. 2f. The lowest concentration of IPTG yields the highest utility. The errors (s.e.m.) are derived from three biological replicates and flow cytometry experiments, each involving 30,000 events. Source data are provided as a Source Data file

behavior of the compensation circuit (Supplementary Fig. 6). At high concentrations of paraquat, the raw mCherry error was positive and increased with $H_2O_2$. Thus, as a first step, we designed an "analog compensation" circuit that increases mCherry expression at high $H_2O_2$ concentrations to compensate for the effects seen in Fig. 3b by expressing *mCherry* under the control of an oxySp promoter (Fig. 3c). The total mCherry levels should be the sum of mCherry expression from the oxySp and pLsoxS promoters (Supplementary Fig. 12B)[30]. This circuit compensated for the $H_2O_2$ crosstalk at high paraquat concentrations, but also increased crosstalk between $H_2O_2$ and *mCherry* expression at low paraquat concentrations (Fig. 3d, Supplementary Fig. 8). Overall, the total relative error of the paraquat-sensing circuit was reduced to 21.2 (Fig. 3h) without significantly affecting the error of the $H_2O_2$-sensing circuit (Fig. 3i).

To address the increased crosstalk at low paraquat concentrations introduced by the "analog compensation circuit", we created a "variable-analog compensation circuit" that was more active at high paraquat concentrations (Fig. 3e, Supplementary Fig. 12C). We fused the C-terminus of the oxySp-regulated mCherry to a TEV protease recognition sequence (TEVrs) and an LAA degradation tag[49]. In the absence of TEV protease (TEVp), the mCherry protein should be rapidly degraded, whereas in the presence of TEVp, the LAA tag should be cleaved off and mCherry post-translationally stabilized[50]. We placed the *TEVp* gene under the control of pLsoxS so that mCherry expressed from oxySp would be unstable unless paraquat induced the expression of *TEVp* (Supplementary Fig. 9). Indeed, increasing paraquat

concentration controlled the magnitude of the mCherry-TEVrs-LAA output from oxySp (Supplementary Fig. 10). This approach considerably reduced the crosstalk at low paraquat concentrations introduced by the analog compensation circuit while maintaining crosstalk compensation at high paraquat concentrations (Fig. 3f, Supplementary Fig. 11). The total relative error was reduced to 18.1 (Fig. 3h) without significantly affecting the error of the $H_2O_2$-sensing circuit (Fig. 3i). Crosstalk was especially reduced for $H_2O_2$ concentrations up to 0.36 mM. At 1.08 mM $H_2O_2$, cell growth was significantly impaired (Supplementary Fig. 13), which likely affected both sensor and crosstalk-compensation circuit functionality.

## Discussion

In this work, we engineered a panel of ROS-sensing circuits and evaluated them with quantitative metrics to determine their analog performance. The utility metric can help to compare analog circuits and, depending on the target application for the circuits, may be useful to bias the utility metric towards the output fold-induction or input dynamic range. For instance, circuits engineered to enable bacteria to detect the level of inflammation in the mammalian gut may prioritize wide dynamic range sensing if the circuit is used to record quantitative memory[51]. Alternatively, circuits engineered to enable bacteria to treat inflammation by producing an anti-inflammatory compound may prioritize a large output fold-induction to amplify drug production. Furthermore, inflammation-sensing bacteria would likely utilize multiple inputs to assess their environment and be

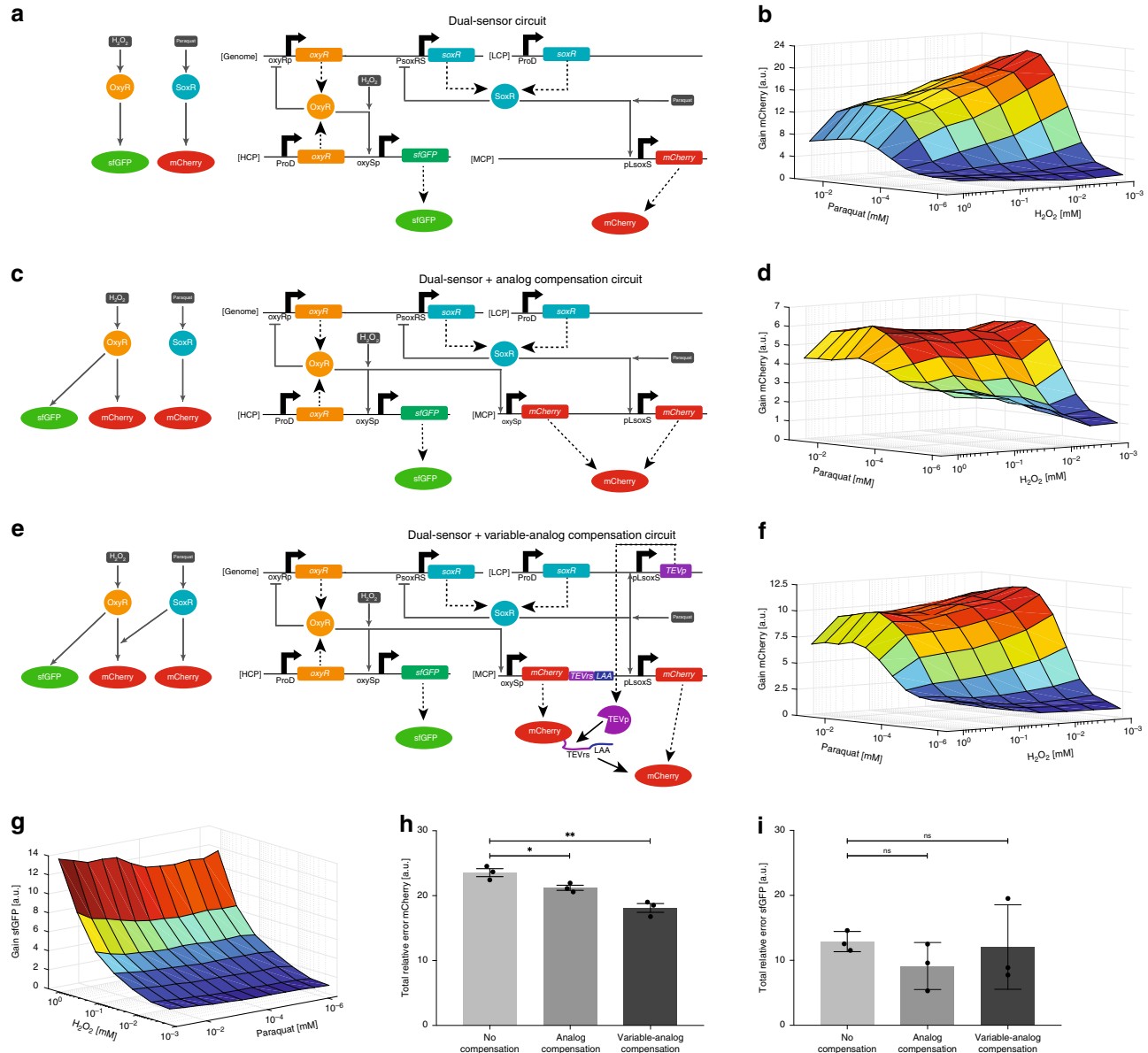

**Fig. 3** Synthetic network-level crosstalk compensation. **a** The first iteration of the dual-sensor circuit. SoxR is constitutively expressed from a LCP and activates *mCherry* expression from pLsoxS on a MCP. OxyR is constitutively expressed from a HCP and activates *sfGFP* expression from oxySp on the same HCP. Genomic *soxR* and *oxyR* expression are both negatively auto-regulated. Dashed large arrows represent protein production and solid small gray arrows represent transcriptional regulation. **b** The mCherry output fold-change relative to the minimum fluorescence for the first iteration dual-sensor strain at different concentrations of $H_2O_2$ and paraquat. *mCherry* expression is mostly dependent upon paraquat concentration, but there is considerable crosstalk between $H_2O_2$ and the mCherry output. **c** The dual-sensor circuit with analog compensation. We added an extra copy of *mCherry* controlled by oxySp to the dual-sensor circuit (Fig. 3a) so that the total mCherry level is controlled by both oxySp and pLsoxS promoter activity. **d** The mCherry output from the dual-sensor circuit with analog compensation. The crosstalk-compensation circuit reduced $H_2O_2$ crosstalk at high paraquat levels, but increased crosstalk at low paraquat levels. **e** The dual-sensor circuit with variable-analog compensation. To the first iteration of the dual-sensor circuit (Fig. 3a), we added an oxySp promoter on a MCP to express mCherry fused to TEVrs and an LAA degradation tag, as well as the *TEVp* gene on a LCP controlled by pLsoxS. TEVp post-translationally cleaves the LAA degradation sequence from the oxySp-expressed mCherry protein at the TEVrs site, stabilizing the mCherry protein. Solid black arrows represent post-translational events. **f** The mCherry output from the dual-sensor circuit with variable-analog compensation. $H_2O_2$ crosstalk was substantially reduced at high paraquat concentrations compared to the original dual-sensor circuit without considerably increasing crosstalk at low paraquat concentrations. **g** The sfGFP output fold-change relative to the minimum fluorescence for the first iteration of the dual-sensor strain (Fig. 3a). *sfGFP* expression is dependent upon $H_2O_2$ concentration and there is little crosstalk with paraquat. **h, i** The total relative mCherry (**h**) and sfGFP (**i**) error. The analog and variable-analog compensation circuits both reduce mCherry crosstalk significantly according to an unpaired two-sided *t*-test, without affecting the sfGFP error. The mean is shown for the total relative error calculated for each experimental replicate individually (represented by the dots). The data in Figs. 3b, d, f and g and the errors (s.e.m.) in Figs. 3h and i are derived from three biological replicates and flow cytometry experiments, each involving 30,000 events. *$p < 0.05$, **$p < 0.005$, ns indicates $p > .05$. In Figs. 3b, d, f, and g, the lowest concentrations of paraquat and $H_2O_2$ tested were zero, but are plotted as non-zero numbers so as to be shown on the logarithmic axes. Source data are provided as a Source Data file

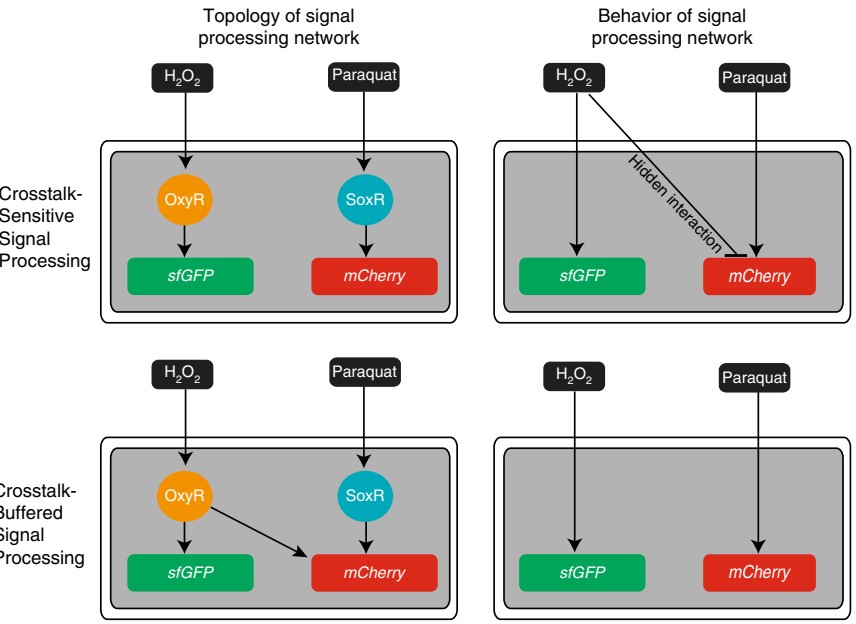

**Fig. 4** The topology of signal-processing networks as perceived from transcription factor-DNA interactions (left column) is contrasted with the actual behavior of these networks (right column). The top row illustrates an example of crosstalk due to a hidden interaction between the inputs and the network. The bottom row illustrates how crosstalk between inputs and the network can be buffered by to crosstalk-compensating circuits

exposed to other off-target molecules that could stimulate their sensor circuits in undesirable ways. Thus, it is essential to quantitatively determine the specificity of circuits for sensing their cognate inputs. The error metrics that we present can be used to quantify crosstalk and guide crosstalk compensation in the future, for example as objective functions for computational circuit design[52–54].

This work illustrates the challenge of building synthetic gene networks with robust behavior in the context of natural regulatory networks and metabolic pathways[55–57], where direct or "hidden" interactions can easily arise between circuits[58,59]. In natural systems, bacterial cells acquire the ability to sense new inputs through the duplication and modification of existing two-component systems[60]. Immediately post-duplication, the histidine kinase and response regulator are expressed at higher levels, leading to decreased performance of the two-component system and to a fitness disadvantage. Therefore, histidine kinase and response regulator undergo high mutation rate at the interface residues to decouple the duplicated histidine kinase from the old response regulator and vice versa[61]. This is followed by changes in the input and output domains, for example through extensive domain swapping, to establish a new sensor pathway.

Rather than trying to address crosstalk by identifying and altering the responsible interactions, we compensated for crosstalk by introducing a new gene network connection (Fig. 4). Consequently, total output protein expression was determined by the sum of two input-processing circuits whose outputs offset each other. Thus, we show that by integrating pathway signals, rather than insulating pathways from each other, synthetic gene networks can be engineered to compensate for crosstalk. This strategy may be especially useful in constructing synthetic networks in contexts where it may be challenging or undesirable to modify endogenous DNA, such as for cell engineering applications[62,63]. We note that our synthetic crosstalk-compensation strategy did not leverage the integration of multiple transcription factors on a single promoter because the rules governing the interaction of multiple transcription factors at a single promoter are not thoroughly understood, even though this may be a viable approach for future signal-integrating designs[64–66].

Our crosstalk compensation circuit forms an incoherent feed-forward loop[67] in its regulation of mCherry when empirically observed interactions are accounted for (Supplementary Fig. 14). It is possible that similar crosstalk-compensating motifs exist in naturally evolved signaling networks to reduce pathway crosstalk via a network connection that is inversely proportional in the undesired activity. These network motifs could be implemented across multiple layers of regulatory cascades. It may be challenging to identify natural crosstalk compensation circuits using computational methods alone and such effects may be difficult to observe in wild-type networks that have evolved to reduce crosstalk; instead, deleting network connections and assaying gene expression in the presence of multiple inputs may reveal network connections that exist to buffer against crosstalk. Altogether, this raises the intriguing idea that crosstalk compensation could be an emergent property arising from the vast interconnectedness[68] of gene regulatory networks. In summary, the analog nature of multi-input synthetic gene circuits can be optimized through quantitative analysis and compensation for undesirable features, such as crosstalk, via network-level signal integration.

## Methods

**Strains and plasmids**. All plasmids were constructed with standard cloning procedures. Parts and plasmids used in this study are detailed in Supplementary Fig. 15 and Supplementary Tables 1 and 2. *Escherichia coli* BW25113 (F⁻, DE(araD-araB)567, lacZ4787(del)::rrnB-3, LAM⁻, rph-1, DE(rhaD-rhaB)568, hsdR514) was used for all experiments unless otherwise noted. *Escherichia coli* MG1655Pro (F⁻ λ⁻ ilvG- rfb-50 rph-1 lacIQ tetR specR) was used for experiments in Figs. 2e, f, g and Supplementary Figs. 4 and 5B. Plasmid sequences and plasmid DNA can be obtained at Addgene under Addgene ID numbers 132571-132586.

**Circuit characterization**. Overnight cultures of *E. coli* were grown from glycerol freezer stocks, shaking aerobically at 37 °C in LB medium with appropriate antibiotics: Carbenicillin (50 μg/ml), Kanamycin (30 μg/ml), Spectinomycin (100 μg/ml). Overnight cultures were diluted 1:100 into fresh LB with antibiotics and grown 1.5 h to an optical density at 600 nm between 0.2 and 0.4. The cell density was adjusted to 50,000 cells/μl and resuspended in Opti-MEM Media + 5% FBS (Invitrogen). 200 μL of culture was transferred to a 96-well plate and inducers were added at appropriate concentrations via serial dilution. The inducers H₂O₂ (H1009-100 ML) and paraquat (methyl viologen dichloride hydrate, 856177-1G) were purchased from Sigma Aldrich. Induced cultures were grown for 1 h shaking

aerobically at 37 °C. Cultures were then diluted 1:4 into a new 96-well plate containing PBS and assayed on a BD LSRFortessa using the high-throughput sampler. At least 30,000 events were recorded for all circuit characterization experiments. Experiments were repeated with three total biological replicates (different colonies obtained from co-transformation of plasmids). sfGFP expression was measured via the FITC channel and mCherry expression was measured via the Texas Red channel. FCS files were exported and processed in FlowJo software. Events were gated for live *E. coli* via forward scatter area and side scatter area, and the geometric mean of the population was calculated. The standard error of the mean (s.e.m.) was calculated from the geometric means of the three biological replicates.

**Reporting summary**. Further information on research design is available in the Nature Research Reporting Summary linked to this article.

## Data availability
The data that support the findings of this study are available from the corresponding author upon request. The source data for Figs. 1, 2, 3, and Supplementary Figs. 3, 9, 10 and 13 are provided as a Source Data file. Plasmids used in this work have been deposited into the Addgene repository under Addgene ID numbers 132571-132586.

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

## Acknowledgements

We would like to thank members of the Lu Lab, the MIT Microbiology Program, and the MIT Synthetic Biology Center for their feedback. We thank the staff at the Koch Institute Flow Cytometry Core for their assistance in flow cytometry. J.R.R. was supported by an NSF Graduate Research Fellowship. This work was supported by the National Science Foundation (#1350625 and #1124247) and the Office of Naval Research (#N000141310424), an NIH New Innovator Award (#1DP2OD008435), and the NIH National Centers for Systems Biology (#1P50GM098792).

## Author contributions

I.E.M., J.R.R. and T.K.L. conceived the study. I.E.M., J.R.R. and T.J. performed experiments and collected data. I.E.M., J.R.R., T.J., D.G., R.X. and T.K.L. analyzed the data and discussed results. I.E.M., J.R.R. and T.K.L. wrote the manuscript.

## Additional information

**Competing interests:** MIT has filed a provisional patent application (PCT/US2015/057478 ('Correcting Crosstalk in Biological Systems')) with I.E.M., J.R.R. and T.K.L. as the named inventors based on this work. All other authors declare no completing interests.

