## [Peer Review File · Nature Communications]

Reviewers' Comments:

Reviewer #1:

Remarks to the Author:

Comments to the authors:

In this manuscript, the authors devise and explore the novel concept of crosstalk compensation in synthetic sensory circuitry. Specifically, the authors show that un-wanted crosstalk in synthetic sensory circuitry can be corrected for by designing counter-crosstalk connections. This is a very interesting strategy, as it tackles cross-talk at the network design level, not at the molecular engineering level (to design regulators that don't show cross-talk in the first place, which has taken up the majority of efforts in synthetic biology to date).

The authors explore this concept through systematically constructing a dual-sensor circuit that utilizes the OxyR and SoxR regulators that sense reactive oxygen species in the cell. They first develop individual sensory circuits with one regulator at a time, and choose the best circuit architecture using a new metric of circuit performance that they introduce called 'utility', which is the product of the fold induction and relative input range. When they combine the OxyR and SoxR circuits they find that OxyR unexpectedly represses the output of the SoxR circuit. Without determining the mechanistic origin of this crosstalk, they find that they can correct for it by introducing additional OxyR activation and selective degradation of the output at low H₂O₂ concentration. In conclusion, they speculate that crosstalk buffering could be used to optimize synthetic gene networks in the context of endogenous systems.

As the field of synthetic biology moves towards designing increasingly sophisticated circuits, concepts like crosstalk correction, introduced in this article, will be vital for maintaining circuit robustness. Natural crosstalk is a hurdle to implementing gene circuits in many systems and the ideas discussed by the authors here could alleviate some of those problems. However, the manuscript is missing some important analyses that would make the concepts relevant to other systems. For example, could the authors construct toy circuits with known levels of cross talk (or even analyzed a minimal version of the ROS sensors in a more controlled manner - i.e. no genomic copies) to determine a quantitative connection between how much extra cross-talk needs to be introduced to correct for a given initial cross-talk? This would be an important point for generalization of the concepts. Another possibility would be to expand upon potential crosstalk compensation in endogenous networks. Can any crosstalk compensation be readily identified from known biological networks?

In terms of specific comments, I have two points that should be addressed before publication:

Introduction - As currently written the introduction feels short and inadequate. The authors could consider adding a systems biology analysis of crosstalk to put their work into a larger context.

Supplementary section 6 - This section is only briefly referenced in the text with very little explanation. It seems like quite a large experiment and should either be given a full explanation in the main text or taken out of the supplementary section entirely.

Reviewer #2:

Remarks to the Author:

The manuscript by Muller et al. entitled "Gene networks that compensate for crosstalk with crosstalk" brought up a new concept that crosstalk can be introduced to compensate the unexpected crosstalk during the engineering and test of synthetic gene networks. The authors investigated two sets of ROS in *E. coli* cells and found H₂O₂ in OxyR system could inhibit mCherry expression in SoxR system in a dose-dependent way, when the two presented in the same cells. Through design of a small additive gene motif to the original circuit, they successfully reduced the

effect of crosstalk in the sensing of the two sets of ROS. Also, they used a metric, "utility", to quantify the fold change and induction range of the two ROS, and quantify the degree of crosstalk.

Given that crosstalk is a widespread "fact" in complicated biological systems and has many different types, this work's crosstalk-compensation strategy is intriguing and will be useful in broader efforts in engineering of synthetic gene circuits. I recommend the publication of the work with some moderate revisions and clarifications as detailed below.

1) The authors employed a concept "utility" to quantify and evaluate a sensor circuit through multiple the relative input range and output fold induction. I am wondering whether "utility" incorporates the nonlinearity (hill coefficient) of inductions, which is also a very important factor for biological systems.

2) In Page 4, line 116. The authors tested OxyR in a positive feedback (PF) motif by fusing mCherry to the C-terminal of OxyR, and the results clearly showed that the PF circuit has a much lower output induction and lower utility than the open loop circuit. Is it because of the fusion influenced mCherry expression levels? Or mCherry influenced OxyR's functionality to activate oxySp promoter and inhibit oxyRp promoter, which finally decreased the PF circuit utility.

3) In Figure 3, the authors found Paraquat has little impact on GFP expression, while H₂O₂ could significantly inhibit mCherry expression. I am wondering why it happens. Is it because of H₂O₂ promoted the stress response and influenced the cell growth? Also, the authors used different copies of plasmids together, whether the heterogeneous gene expression influences cell growth and output ranges. Grow curve assay may be needed for better interpretation of the data.

4) The compensation circuit in Figure 3d and 3f function well and reduce the crosstalk. However, it is possible that the observed mCherry expression is mainly from oxySp promoter (induced by H₂O₂), but not pLsoxS promoter (induced by Paraquat). Especially, Paraquat has small output fold range in Figure 3e. The authors may need test a control circuit without soxR-pLsoxS-mCherry parts. Similar problems for Figure 3f.

A minor point: in Figure 3c, 3e, 3g, it looks like the dot for each dosage only has one data point, does the data point is an averaged one from replicates? For reader convenience, the authors may need to add the information in the figure legends.

Point-By-Point Response to Reviewers' Comments

Reviewer #1

Comments to the authors:

In this manuscript, the authors devise and explore the novel concept of crosstalk compensation in synthetic sensory circuitry. Specifically, the authors show that unwanted crosstalk in synthetic sensory circuitry can be corrected for by designing counter-crosstalk connections. This is a very interesting strategy, as it tackles cross-talk at the network design level, not at the molecular engineering level (to design regulators that don't show cross-talk in the first place, which has taken up the majority of efforts in synthetic biology to date).

The authors explore this concept through systematically constructing a dual-sensor circuit that utilizes the OxyR and SoxR regulators that sense reactive oxygen species in the cell. They first develop individual sensory circuits with one regulator at a time, and choose the best circuit architecture using a new metric of circuit performance that they introduce called 'utility', which is the product of the fold induction and relative input range. When they combine the OxyR and SoxR circuits they find that OxyR unexpectedly represses the output of the SoxR circuit. Without determining the mechanistic origin of this crosstalk, they find that they can correct for it by introducing additional OxyR activation and selective degradation of the output at low H₂O₂ concentration. In conclusion, they speculate that crosstalk buffering could be used to optimize synthetic gene networks in the context of endogenous systems.

As the field of synthetic biology moves towards designing increasingly sophisticated circuits, concepts like crosstalk correction, introduced in this article, will be vital for maintaining circuit robustness. Natural crosstalk is a hurdle to implementing gene circuits in many systems and the ideas discussed by the authors here could alleviate some of those problems. However, the manuscript is missing some important analyses that would make the concepts relevant to other systems. For example, could the authors construct toy circuits with known levels of cross talk (or even analyzed a minimal version of the ROS sensors in a more controlled manner - i.e. no genomic copies) to determine a quantitative connection between how much extra cross-talk needs to be introduced to correct for a given initial cross-talk? This would be an important point for generalization of the concepts. Another possibility would be to expand upon potential crosstalk compensation in endogenous networks. Can any crosstalk compensation be readily identified from known biological networks?

Thanks for your comments and suggestions, we appreciate your effort to help us improve our manuscript.

In our opinion it might be challenging to precisely model and predict what levels of crosstalk need to be introduced for compensation, as this will strongly depend on the observed crosstalk system (e.g. inputs, outputs, transcription factors, promoter, RBSs and vector copy number). Hence, we suggest to perform a stepwise crosstalk compensation as done in the manuscript. First, researchers determine the level of counter crosstalk that is necessary to compensate crosstalk by tuning plasmid copy number and RBS strength. In the next step, dependence with the 2nd inducer is introduced to finalize the crosstalk compensated circuit. We see this work as proof of principle of a new approach on how to compensate crosstalk, and not so much a model.

To address the final point, we expanded both the introduction and discussion with a brief paragraph about natural crosstalk occurring in two-component-systems and how cells insulate signaling pathways from each other to acquire the ability to sense a new input. It is important to keep in mind that crosstalk in endogenous systems often leads to a fitness disadvantage and is therefore selected against. On the other hand, if a synthetic system is introduced to perform computation within cells, unwanted crosstalk can occur. We therefore think that it would be hard to predict crosstalk in synthetic gene networks and the strategy presented here can serve as a roadmap to correct unwanted responses.

In terms of specific comments, I have two points that should be addressed before publication:

- 1) Introduction - As currently written the introduction feels short and inadequate. The authors could consider adding a systems biology analysis of crosstalk to put their work into a larger context.*

Thanks for the comment. As pointed out in the response above, we expanded the introduction and discussion with a small paragraph covering crosstalk occurring in two-component-systems. We hope this will put our work in a larger context and emphasizes the importance of crosstalk compensation approaches.

- 2) Supplementary section 6 - This section is only briefly referenced in the text with very little explanation. It seems like quite a large experiment and should either be given a full explanation in the main text or taken out of the supplementary section entirely.*

We agree that the supplementary section is not an adequate place for this experiment, but think that it would distract too much from the main point of the paper if added to the main text. We decided to remove it entirely.

Reviewer #2

The manuscript by Muller et al. entitled “Gene networks that compensate for crosstalk with crosstalk” brought up a new concept that crosstalk can be introduced to compensate the unexpected crosstalk during the engineering and test of synthetic gene networks. The authors investigated two sets of ROS in E. coli cells and found H₂O₂ in OxyR system could inhibit mCherry expression in SoxR system in a dose-dependent way, when the two presented in the same cells. Through design of a small additive gene motif to the original circuit, they successfully reduced the effect of crosstalk in the sensing of the two sets of ROS. Also, they used a metric, “utility”, to quantify the fold change and induction range of the two ROS, and quantify the degree of crosstalk.

Given that crosstalk is a widespread “fact” in complicated biological systems and has many different types, this work’s crosstalk-compensation strategy is intriguing and will be useful in broader efforts in engineering of synthetic gene circuits. I recommend the publication of the work with some moderate revisions and clarifications as detailed below.

Thank you for your time and work to bring these points to our attention.

- 1) The authors employed a concept “utility” to quantify and evaluate a sensor circuit through multiple the relative input range and output fold induction. I am wondering whether “utility” incorporates the nonlinearity (hill coefficient) of inductions, which is also a very important factor for biological systems.*

Thanks for the suggestion. While the output fold induction is independent of the nonlinearity, the relative input range (and therefore the utility) depends on the hill coefficient. We expanded the calculations in the supplementary section 1 to point out the relation between the relative input range and the hill coefficient.

- 2) In Page 4, line 116. The authors tested OxyR in a positive feedback (PF) motif by fusing mCherry to the C-terminal of OxyR, and the results clearly showed that the PF circuit has a much lower output induction and lower utility than the open loop circuit. Is it because of the fusion influenced mCherry expression levels? Or mCherry influenced OxyR’s functionality to activate oxySp promoter and inhibit oxyRp promoter, which finally decreased the PF circuit utility.*

We hypothesize that the mCherry fusion influences OxyR’s ability to bind to the oxyS promoter. OxyR binds as a tetramer and a protein fusion could prevent proper structure formation due to space constraints. It is also possible that protein levels of OxyR are lower, as the proD promoter is a strong, constitutive promoter. Due to the decreased performance of the PF circuit, we moved forward with the OL design.

- 3) *In Figure 3, the authors found Paraquat has little impact on GFP expression, while H₂O₂ could significantly inhibit mCherry expression. I am wondering why it happens. Is it because of H₂O₂ promoted the stress response and influenced the cell growth? Also, the authors used different copies of plasmids together, whether the heterogeneous gene expression influences cell growth and output ranges. Grow curve assay may be needed for better interpretation of the data.*

Thanks for this suggestion. In Fig. S14 we include growth curves for the dual-sensor circuit in Fig. 3A for the two highest H₂O₂ concentrations (1.08mM and 0.36mM) and titrating paraquat concentrations. We believe that the strong reduction in mCherry signal (Fig. 3C) at 1.08mM H₂O₂ can be explained by cell growth inhibition. We did not further explore other possible causes for the crosstalk behavior, as the advantage of our crosstalk compensation approach is that the underlying cause does not need to be investigated.

- 4) *The compensation circuit in Figure 3d and 3f function well and reduce the crosstalk. However, it is possible that the observed mCherry expression is mainly from oxySp promoter (induced by H₂O₂), but not pLsoxS promoter (induced by Paraquat). Especially, Paraquat has small output fold range in Figure 3e. The authors may need test a control circuit without soxR-pLsoxS-mCherry parts. Similar problems for Figure 3f.*

Thanks for raising this point. The decreased output fold range in Fig. 3E can be explained by higher basal expression levels of mCherry in comparison to Fig. 3C. The basal expression level in Fig. 3G is reduced as mCherry is fused to a degradation tag.

The circuit in Fig. S10A is the same circuit as shown in Fig. 3F, but lacks the paraquat induced TEVp. Therefore, H₂O₂ induced mCherry should get degraded efficiently as the degradation tag cannot be cleaved off. The transfer function shown in Fig. S10B is very similar to Fig. 3C, demonstrating that the H₂O₂ induced mCherry only contributes to the crosstalk compensation circuit if the degradation tag is removed by TEVp.

The circuit in Fig. S11A lacks paraquat induced mCherry. Here, the mCherry gain is very high, in particular for high amounts of H₂O₂ and paraquat. However, this high mCherry gain originates mainly from the very low basal expression level at 0mM H₂O₂ and 0mM paraquat. As discussed previously, the basal expression of H₂O₂ induced mCherry is so low, as it gets efficiently degraded in the absence of TEVp. It is hard to compare Fig. S11B to Fig. 3G due to significantly different mCherry basal expression levels, but it seems clear that the paraquat induced mCherry transfer function originates from paraquat-soxR induced mCherry, and not from H₂O₂ induced mCherry.

5) *A minor point: in Figure 3c, 3e, 3g, it looks like the dot for each dosage only has one data point, does the data point is an averaged one from replicates? For reader convenience, the authors may need to add the information in the figure legends.*

Thanks for the comment. The data points are averaged from three biological replicates. We added a sentence in the figure captions for clarification.

Reviewers' Comments:

Reviewer #1:

Remarks to the Author:

We appreciate the authors' response to the reviewers comments. We can appreciate that the additional studies previously requested may be out of the scope of this work. Nevertheless there are many very interesting ideas in this manuscript that would be appreciated by the readership of Nature Communications, and we therefore think this paper is suitable for publication.

Reviewer #2:

Remarks to the Author:

In the manuscript titled "Gene networks that compensate for crosstalk with crosstalk", Muller et al. constructed versions of synthetic circuit with the aim to compensate for signal crosstalk and hence increase the systems' sensitivity and specificity for certain signals, which the authors quantified as the newly defined "utility". The concept is borrowed from electrical engineering and the authors indeed showed significant increase of utility of the pathways. After reading the manuscript and reviewers' comments along with authors' response, I believe the authors fully and sufficiently addressed reviewers' concerns. And I have no further concerns other than already brought up by 1st round reviewers. Therefore, I would recommend for its publication without reservation.

Point-By-Point Response to Reviewer's Comments

Reviewer #1

We appreciate the authors' response to the reviewers comments. We can appreciate that the additional studies previously requested may be out of the scope of this work. Nevertheless there are many very interesting ideas in this manuscript that would be appreciated by the readership of Nature Communications, and we therefore think this paper is suitable for publication.

Thank you for your time and comments on our manuscript.

Reviewer #2

In the manuscript titled "Gene networks that compensate for crosstalk with crosstalk", Muller et al. constructed versions of synthetic circuit with the aim to compensate for signal crosstalk and hence increase the systems' sensitivity and specificity for certain signals, which the authors quantified as the newly defined "utility". The concept is borrowed from electrical engineering and the authors indeed showed significant increase of utility of the pathways. After reading the manuscript and reviewers' comments along with authors' response, I believe the authors fully and sufficiently addressed reviewers' concerns. And I have no further concerns other than already brought up by 1st round reviewers. Therefore, I would recommend for its publication without reservation.

We thank the reviewer for taking the time to read the manuscript, the reviewer's comments and our reply.